# Layered Unlearning for Adversarial Relearning

## Abstract

Our goal is to understand how post-training methods, such as fine-tuning, alignment, and unlearning, modify language model behavior and representations. We are particularly interested in the brittle nature of these modifications that makes them easy to bypass through prompt engineering or relearning. Recent results suggest that post-training induces shallow context-dependent "circuits" that suppress specific response patterns. This could be one explanation for the brittleness of post-training. To test this hypothesis, we design an unlearning algorithm, Layered Unlearning (LU), that creates distinct inhibitory mechanisms for a growing subset of the data. By unlearning the first $i$ folds while retaining the remaining $k - i$ at the $i$th of $k$ stages, LU limits the ability of relearning on a subset of data to recover the full dataset. We evaluate LU through a combination of synthetic and large language model (LLM) experiments. We find that LU improves robustness to adversarial relearning for several different unlearning methods. Our results contribute to the state-of-the-art of machine unlearning and provide insight into the effect of post-training updates.

## 1 Introduction

Post-training interventions such as fine-tuning, preference learning, and unlearning are widely used to modify the behavior of pre-trained large language models (LLMs). However, changes introduced in post-training are often brittle. However, these changes are often shallow or brittle. In many cases, they are bypassed or reversed by clever adversarial prompting or fine-tuning (Jain et al., 2024; Arditi et al., 2024; Zou et al., 2023; Greenblatt et al., 2024; Che et al., 2024; Deeb & Roger, 2025; Betley et al., 2025). Our goal is to understand how post-training methods modify language model behavior and representation and support the design of more robust post-training methods.

We study this through the lens of machine unlearning, which seeks to remove knowledge or capabilities from pre-trained models. Deeb & Roger (2025) recently demonstrated that "unlearned" information is easily re-elicited by fine-tuning on a subset of the removed data. To explain this result, we hypothesize that SoTA unlearning methods introduce a context-dependent *inhibitor* mechanism. Efficient "relearning" generalizes because fine-tuning removes a single shared mechanism and reverses the full post-training modification.

A common way to mitigate single failure points is the famous "Swiss cheese" defense-in-depth model (Reason, 1990). The goal is to combine multiple imperfect defenses. If their failure modes are distinct, the combined defense is more robust than any individual approach. We implement defense-in-depth through **Layered Unlearning** (LU). LU partitions the data into $k$ disjoint folds and applies unlearning sequentially to a growing subset: at stage $i$, LU unlearns the union of folds $F_1$ through $F_i$. Crucially, we retain the data from $F_{i+1}$ through $F_k$ to induce distinct inhibitors at each stage of unlearning. Figure 1 illustrates the algorithm and the defense-in-depth inspiration.

We investigate the performance of LU on a variety of synthetic tasks and unlearning benchmarks. We consider a synthetic 2-dimensional classification task and a 3-token sequence generation task. Next, we apply LU to a variety of unlearning methods on the WMDP, Years, and MMLU datasets. In all settings, LU improves

---

Synthetic experiment code at: `https://anonymous.4open.science/r/layered-unlearning-380C`

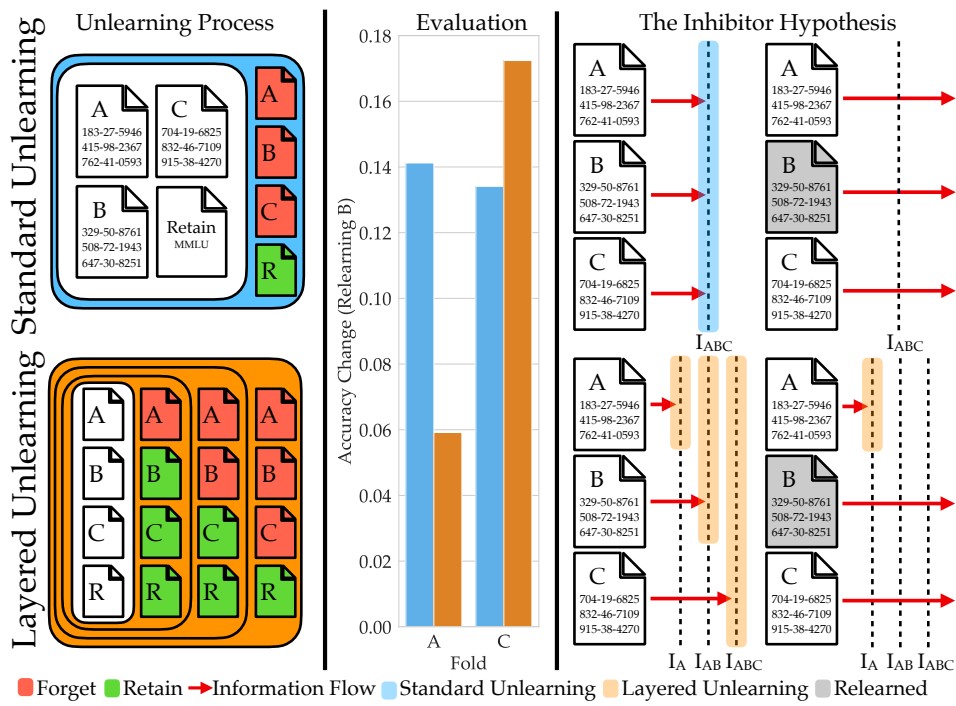

Figure 1: **Left:** An illustration of LU with social security numbers (SSNs). The SSNs are partitioned into disjoint sets $(A, B, C)$. *Top:* Standard unlearning minimizes performance on $A \cup B \cup C$ while retaining general capabilities on a retain set $R$ (e.g., MMLU). *Bottom:* In LU, we sequentially unlearn the sequence $\{A, A \cup B, A \cup B \cup C\}$ while retaining the sequence $\{B \cup C \cup R, C \cup R, R\}$. **Middle:** As a result, relearning $B$ improves performance on $C$ but not $A$. In contrast, training on any subset improves performance across the board for standard methods. **Right:** We hypothesize that unlearning the full set introduces a context-dependent shared *inhibitor* $I_{ABC}$ that suppresses the information and that subsequent relearning removes $I_{ABC}$. The structure of LU is designed to create several distinct inhibitors $I_A, I_{AB}, I_{ABC}$ that cover different folds of the data. Relearning on $B$ removes $I_{AB}$ and $I_{ABC}$, but leaves $I_A$ active.

resistance to fine-tuning-based recovery. In the course of these experiments, we also identify a stronger class of attack: *corpus-based fine-tuning*. This attack, which uses raw text rather than structured MCQ-based prompts, bypasses inhibitors more effectively than standard RTT (Deeb & Roger, 2025). Notably, this distinction only emerges because LU creates variation in robustness across data folds—an effect that is not observed with standard unlearning.

We make three contributions: 1) we introduce *Layered Unlearning* (LU) a method that combines multiple steps of unlearning to increase robustness; 2) we show in both synthetic and LLM settings that LU improves resistance to adversarial relearning; and 3) we use LU to reveal a gap in attack strength between MCQ-based and corpus-based relearning, offering new insight into the limits of post-training behavioral control. Our results contribute to the state-of-the-art of machine unlearning and provide insight into the effect of post-training updates.

## 2 Layered Unlearning

In this section, we introduce LU. We begin with a replication of Deeb & Roger (2025) in two synthetic tasks: a 2D classification task with mixtures of Gaussians and a bigram completion task with three tokens. Next, we introduce the LU algorithm. Finally, we investigate the effect of LU in this task. We find that LU improves robustness in both cases and analyze the sequences of changes that LU induces. First we introduce some notation to represent a machine unlearning method $U$.

## 2.1 Machine unlearning notation

The goal of machine unlearning is to remove information $F$ from trained model weights $\theta \in \Theta$ that model a dataset $D$ in some input space $X$. With unlimited compute, this would involve retraining from scratch on $D \setminus F$. Due to, e.g., the cost of pretraining, unlearning methods attempt to approximate this result as a post-training step that maintains performance on a retain set $R \subset D$.

Thus, we can represent a generic unlearning algorithm $U$ as a function that maps model parameters $\theta$, forget set $F$, retain set $R$, and hyperparameters $\gamma \in \Gamma$ to a new set of model parameters $\theta'$. When clear from context, we may omit the final argument corresponding to the hyperparameters. Formally

$$U : \Theta \times X \times X \times \Gamma \to \Theta.$$

## 2.2 Adversarial relearning in synthetic settings

We replicate the results of Deeb & Roger (2025) in two synthetic settings: a 2D classification task and a bigram language modeling task.

**2D logistic regression.** Our first task is a 2D logistic regression task, so our input space is $X = \mathbb{R}^2$. The goal is to classify a mixture of Gaussians (class 1) against a uniform background distribution over $\mathcal{U}([-60, 60]^2)$ (class 0). We sample the Gaussian means from the uniform distribution $\mathcal{U}([-50, 50]^2)$ and use a primarily isotropic covariance matrix with variance $\sigma^2 = 4$, adding small perturbations of magnitude 0.1 to break exact symmetry. We implement a linear classifier with logistic regression on radial basis functions (RBF) features.

We partition the Gaussians into subsets $A, B, R$, where the goal is to unlearn $A \cup B$ and retain $R$. Tasks $A$ and $B$ are defined as the classification accuracy when sets $A$ and $B$ are labeled as class 1, and the retain task as the joint accuracy on classifying $R$ as class 1 and Null as class 0. We place the RBF centers on an $12 \times 12$ grid of points, so $\Theta = \mathbb{R}^{145}$ (including a bias term). The top left of figure 2 shows the learned weights with 2 folds.

The unlearning primitive $U$ takes as input model parameters $\theta$, a forget set $F$, a retain set $R$, and hyperparameters $\gamma$. Each data point in $F$ and $R$ is a 2D input. The objective is to preserve the original classification for points in $R$, while reclassifying points in $F$ as class 0. Relearning refers to assigning data points in the relearned set back to their original classifications. We optimize all objectives using the Adam optimizer.

**Bigram sequence modeling.** Next, we consider a bigram language modeling task with three tokens: $a, b, r$. The input space is length 8 token sequences: $X = \{a, b, r\}^8$. We generate data so that $a$ and $b$ are followed by $r$, while $r$ is followed by $a$ and $b$ with equal probability. We combine this with a small minimum probability $\epsilon = 0.05$ of a uniform transition across tokens to encourage smooth learning dynamics. This leads to the following conditional probabilities for consecutive tokens: $P(r \mid a) = P(r \mid b) = 1 - 2\epsilon$, $P(a \mid r) = P(b \mid r) = \frac{1}{2} - \frac{\epsilon}{2}$, and $P(a \mid a) = P(b \mid a) = P(a \mid b) = P(b \mid b) = P(r \mid r) = \epsilon$.

We define tasks $A$, $B$, and $R$ as the prediction performance on all consecutive pairs tokens in the bigram sets $\{aa, ab, ar\}$, $\{ba, bb, br\}$, and $\{ra, rb, rr\}$, respectively. We use a small attention-only, one-layer transformer with parameter space $\Theta = \mathbb{R}^{4288}$. This architecture is simple enough to permit analytical study (Elhage et al., 2021), while still serving as a useful proxy for the larger models used in our later experiments.

The unlearning primitive $U$ takes as input model parameters $\theta$, a forget set $F$, a retain set $R$, and hyperparameters $\gamma$. Data points in $F$ and $R$ are elements of $\{a, b, r\}$. For each $x \in R$, the objective is to preserve the original conditional distribution over consecutive tokens, $P(\cdot \mid x)$. For each $x \in F$, the goal is to flatten the distribution to $P(\cdot \mid x) = \frac{1}{3}$. The retain task prevents the model from collapsing to a trivial uniform distribution over tokens after unlearning. To implement unlearning, we generate sequences from this modified transition matrix and optimize the standard language modeling loss. Relearning refers to restoring the original conditional distribution over tokens for the subset being relearned. We optimize all objectives using the Adam optimizer.

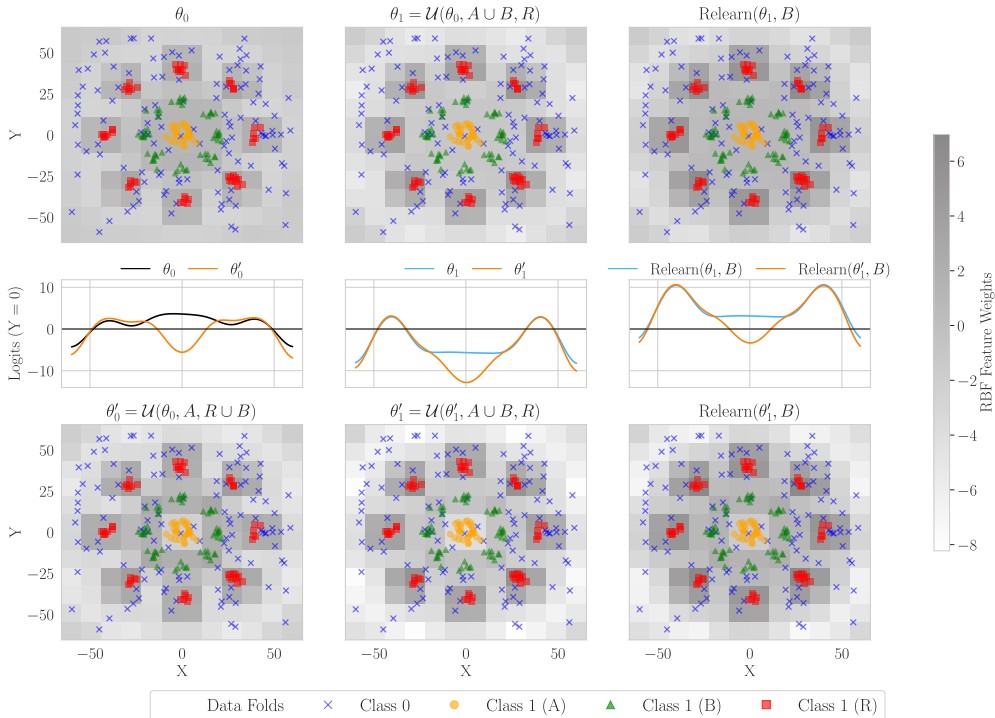

Figure 2: A depiction of Layered Unlearning in our 2D logistic regression (LR) example. Scatter plots represent the data, which consists of a uniform distribution (Class 0) and a mixture of Gaussians (Class 1) that is split into three subsets, $A$, $B$, and $R$. The goal of unlearning is to forget $A, B$ while retaining $R$. Our classifiers are trained with LR with radial basis functions spaced out in a grid. We show the weights as a heatmap on the grid. The top left shows $\theta_0$, the initial trained model. Across the top row, we illustrate the effect of joint unlearning $\theta_1 = (\theta_0, A \cup B, R)$ and subsequent relearning on $B$. Note that relearning $B$ also relearns $A$. The corresponding classification logits along $Y = 0$ are shown below. Notice learning on $B$ generalizes to the area around $(0,0)$. In the bottom row, we show the steps of Layered Unlearning. First, we compute $\theta_0' = U(\theta_0, A, R \cup B)$, shown in the bottom left, then we compute $\theta_1' = U(\theta_0', A \cup B, R)$, shown in the bottom middle. The logit plot shows the clear effect on the logits near $(0,0)$. The bottom right shows the effect of subsequently relearning $B$, performance on $B$ still improves but it no longer generalizes to $A$.

**Relearning performance.** For each task, we first apply the unlearning primitive, followed by relearning on each subtask. We limit our discussion to relearning on $B$ and evaluating on $A$ for symmetry's sake. In 2D classification, relearning on $B$ restores 93% of task $A$'s original performance (Table 1). Figure 2 (top) illustrates the corresponding weight trajectories. In bigram modeling, we observe similar behavior: relearning on $B$ recovers 73% of task $A$'s performance (Table 2).

## 2.3 The Layered Unlearning algorithm

Next, we present the LU algorithm and evaluate it in these two domains. Algorithm 1 shows the algorithm details. The algorithm relies on an unlearning primitive $U$ that maps model weights $\theta$, forget set $F$, retain set $R$, and algorithm hyperpararmeters $\gamma$

Its primary input is a set of model weights $\theta_0$, a sequence of $k$ forget sets $\{F_i\}$, a retain set $R$, an unlearning algorithm $U$, and a sequence of algorithm hyperparameters $\{\gamma_i\}$. LU proceeds through $k$ steps. At step $i$, LU computes $\theta_i = U(\theta_{i-1}, F_1 \cup F_2 \cdots \cup F_i, R \cup F_{i+1} \cdots \cup F_k, \gamma_i)$ : it unlearns $F_1 \cup F_2 \cdots \cup F_i$ while retaining $R \cup F_{i+1} \cdots \cup F_k$.

We analyze LU through inhibitors. In the 2-fold case of unlearning tasks $A$ and $B$, the first stage forgets $A$ while retaining $B$, which forces the model to activate an inhibitor $I_A$ that selectively suppresses performance

on $A$. In the second stage, the model may activate either an inhibitor $I_B$ that targets $B$ specifically, or a shared inhibitor $I_{AB}$ that suppresses both $A$ and $B$.

Upon relearning $B$, any inhibitors affecting $B$—namely $I_B$ or $I_{AB}$—are deactivated, but $I_A$ remains active, so performance on $A$ stays suppressed. Conversely, when relearning $A$, the inhibitors $I_A$ and $I_{AB}$ are deactivated. If $I_{AB}$ had been activated, performance on $B$ is also restored; however, if $I_B$ had been activated instead, performance on $B$ remains suppressed. Whether the barrier to adversarial relearning is unidirectional or bidirectional depends on the unlearning primitive. 3-fold LU is illustrated in Figure 1.

---

**Algorithm 1** Layered Unlearning

---

**Require:** Model parameters $\theta_0$, forget dataset sequence $\{F_1, \ldots, F_k\}$, retain dataset $R_0$, hyperparameters $\{\gamma_1, \ldots, \gamma_k\}$, unlearning algorithm $U$
**Ensure:** Unlearned model $\theta_k$
 1: $F = \emptyset, R = R_0 \bigcup_{i=1,\ldots,k} F_i$          $\triangleright$ Initialize incremental forget and retain sets $F$ and $R$.
 2: **for** $i = 1$ to $k$ **do**          $\triangleright$ Iterate through sequential forget stages
 3:      $F = F \cup F_i, R = R \setminus F_i$          $\triangleright$ Update forget and retain sets
 4:      $\theta_i = U(\theta_{i-1}, F, R, \gamma_i)$          $\triangleright$ Apply unlearning to this fold
 5: **end for**
 6: **return** $\theta_k$

---

### 2.3.1 Layered Unlearning for logistic regression

Table 1: LU performance for logistic regression with random Gaussian assignment and 5 Gaussians per dataset. (10 random seeds average).

| Method | Relearn | A ↓ | B ↓ | R ↑ |
|---|---|---|---|---|
| Original | — | 1.00 | 1.00 | 0.88 |
| U | — | 0.02 | 0.01 | 0.96 |
| U-LU | — | 0.01 | 0.01 | 0.96 |
| U | A | — | **0.93** | **0.80** |
| U-LU | A | — | 0.96 | 0.78 |
| U | B | 0.93 | — | 0.80 |
| U-LU | B | **0.30** | — | **0.86** |

The bottom row of Figure 2 shows the sequence of weights that are generated by 2-fold LU. To illustrate the effect, we arrange $A, B, R$ as concentric circles with $A$ in the center. We can see that the most central weights $(\pm5, \pm5)$ decrease to forget $A$ while the surrounding circle of weights *increases* to retain $B$. This distinction is preserved when $A \cup B$ is unlearned and so relearning $B$ does not recover performance on $A$. With 5 Gaussians in the dataset, relearning on $B$ increases accuracy on $A$ by only 0.29, compared to a 0.91 increase in accuracy on $B$ when relearning on $A$ (see Table 1).

We conducted experiments varying both the number of Gaussians in the mixture and the procedures used to assign the Gaussians to $A, B, R$. We find that increased task overlap leads to more adversarial relearning. This can occur by increasing the number of Gaussians per cluster (expanding the region of potential overlap) or by randomly assigning Gaussians to $A$, $B$, and $R$. To reduce overlap, we also cluster Gaussian means and assign entire clusters to $A$, $B$, and $R$, which significantly reduces adversarial relearning across all unlearning algorithms.

While the setup in Figure 2 is deliberately simplified to visualize inhibitors, we observe similar trends across all configurations (Appendix B). In particular, when Gaussian means are randomly sampled—causing more overlap between $A$, $B$, and the retain task—standard unlearning becomes notably less robust. In contrast, when components are clustered to make folds more distinct, robustness improves, likely due to increased dataset separation making adversarial relearning more difficult.

Table 2: LU performance for bigram language modeling with a 1-layer attention-only transformer. (Results averaged across 10 random seeds).

| Method | Relearn | A ↓ | B ↓ | R ↓ |
|--------|---------|-----|-----|-----|
| Original | — | 0.91 | 0.91 | 0.02 |
| U | — | 0.33 | 0.34 | 0.01 |
| LU | — | 0.34 | 0.33 | 0.02 |
| U | A | — | 0.78 | 0.06 |
| LU | A | — | **0.53** | **0.04** |
| U | B | 0.76 | — | 0.05 |
| LU | B | **0.51** | — | **0.04** |

### 2.3.2 Layered Unlearning for bigram modeling

We measure the prediction performance for tasks $A, B$ with prediction accuracy. For the retain task $R$, we measure the total variation distance from a uniform distribution over $a, b$. We show the performance of the original weights, the unlearned weights with U and LU respectively, and the performance after relearning on $A$ or $B$.

In this case, LU also confers bidirectional robustness to relearning generalization. While relearning $A$ or $B$ after U increases performance on the other task by 0.43 on average. After LU, generalization accuracy only improves by 0.17 on average. In contrast to our other experiments, we find that LU seems to activate fully independent inhibitors so that relearning does not transfer $A \to B$ or $B \to A$ (see Table 2).

To better understand the source of LU's robustness, we conduct ablations on transformer components (see Appendix B). We find that the components from U and LU models are interchangeable without affecting task performance, including on the retain set. However, the attention components—specifically the $QK$ and $OV$ circuits—are essential for resisting adversarial relearning, suggesting that robustness is encoded in the model's attention mechanisms.

Consistent with this, retain set performance remains stable under relearning, indicating that the transformer does not revert to uniform predictions but instead applies targeted inhibition to Tasks $A$ and $B$. In contrast, a zero-layer, embedding-only transformer exhibits little to no adversarial relearning, highlighting the role of depth and attention in shaping inhibitor behavior for this setting. While we do not fully explain this result, we release all code and data to facilitate future interpretability research.

## 3 LLM unlearning experiments

Next, we evaluate the performance of LU on LLM unlearning benchmarks. Specifically, we consider unlearning on the WMDP (Li et al., 2024b), MMLU (Hendrycks et al., 2021), and Years (Deeb & Roger, 2025) datasets. WMDP consists of dangerous knowledge framed as multiple-choice questions (MCQs). To assess LU's ability to remove capability-related information, we also apply unlearning to subsets of MMLU directly. The Years dataset contains major world events annotated with the year in which they occurred. For retain set evaluation, we use MMLU; when unlearning on MMLU, we exclude the categories being unlearned from the retain set.

**Unlearning.** State-of-the-art LLM unlearning methods fall into two categories: representation engineering and gradient ascent. We select a representative algorithm from each for our unlearning primitive $U$: *Representation Misdirection Unlearning* (RMU) (Li et al., 2024a) for representation engineering, and *Simple Negative Policy Optimization* (SimNPO) (Fan et al., 2025) for gradient ascent. We evaluate both methods with and without LU, denoting the LU variants as L-RMU and L-SimNPO, respectively. For a graphical overview of the unlearning process, see Appendix Figure 5.

**Evaluation.** Given a dataset $F$ to be unlearned, we uniformly at random split it into $k$ folds, $F_1, \ldots, F_k$. For a fixed $k$ and dataset, this partitioning remains consistent across all experiments for both unlearning and evaluation. We follow the Language Model Evaluation Harness standards for 0-shot evaluation (Gao et al., 2023).

To evaluate a model $\mathcal{M}$, we consider all $2^k - 2$ proper subsets $S \subset \{F_1, \ldots, F_k\}$ and follow the evaluation protocol in Deeb & Roger (2025). Specifically, we fine-tune $\mathcal{M}$ on either the MCQ data or the corresponding corpus from $S$ and then evaluate it on the MCQ questions from $T := F \setminus S$. We track accuracy on T over epochs and report the best accuracy as the final forget accuracy. All fine-tuning experiments use the Adam optimizer (Kingma & Ba, 2017).

To ensure model utility, we only consider unlearned models that experience at most a 10% accuracy drop on the retain set, see Appendix Table 6. All experiments are conducted using *Zephyr-7B-β* (Tunstall et al., 2023).

### 3.1 Layered Unlearning is more robust to adversarial relearning

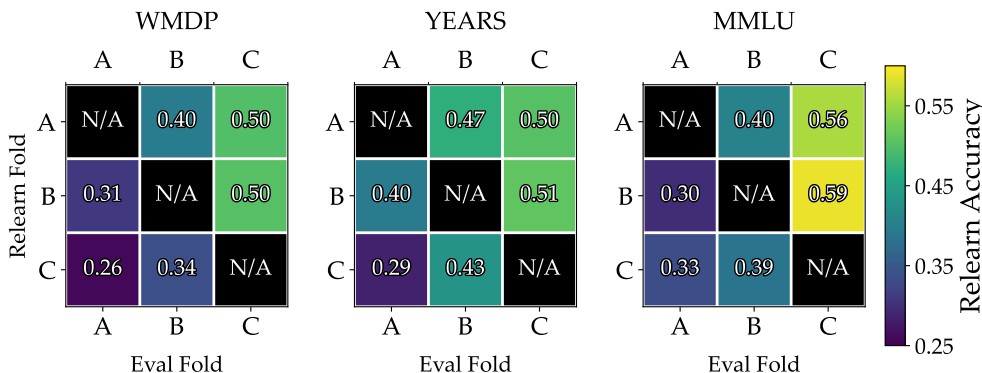

Figure 3: Model accuracy after relearning different folds of the data for an experiment with Layered RMU (L-RMU) and the folds $\{A, B, C\}$ in order. Each row shows the performance per fold for different relearning subsets. Notice that values below the diagonal are lower than values above the diagonal. This shows that L-RMU introduces a one-way barrier to relearning: relearning on $B$ regains performance on $C$ but not on $A$.

Across our experiments, we find that RMU becomes more robust to adversarial relearning when augmented with LU (Figure 3). This robustness is sensitive to the order of folds: an adversary with access to fold $A$ can recover more information about $B$ and $C$ than one with access only to fold $C$. This reflects a path-dependent property of LU, where the sequence of unlearning influences the model's vulnerability to relearning.

SimNPO exhibits a similar, though weaker, improvement in robustness under LU. Notably, L-SimNPO produces a more symmetric barrier effect, in contrast to the directional robustness seen with RMU. These results indicate that LU generalizes across different unlearning methods.

### 3.2 Corpus-based fine-tuning is a stronger adversarial attack

We investigate the limits of LU by replacing MCQ-based prompts with corpus-based fine-tuning. While MCQ-based fine-tuning is commonly used due to its guaranteed performance improvement on targeted questions, corpus-based fine-tuning may more directly realign internal representations, making it a potentially stronger attack—particularly against unlearning methods based on representation engineering.

This substitution reveals a new state-of-the-art attack—one that only becomes apparent because LU enhances robustness to standard MCQ-based attacks. Although some robustness remains, it is reduced, as shown in Figure 3 and Table 3. Concretely, RMU's performance increases by an average of 5% under corpus-based fine-tuning. For L-RMU, the effect is larger: performance improves by 10% on average when relearning on

later folds and evaluating on earlier ones, and by 6% when relearning on earlier folds and evaluating on later ones. This asymmetry emerges only because LU introduces additional robustness, revealing corpus-based fine-tuning as a more effective attack in certain cases.

Interestingly, both SimNPO and L-SimNPO are less affected by corpus-based fine-tuning. However, SimNPO remains more vulnerable to adversarial relearning overall. This contrast shows that unlearning methods differ not only in their effectiveness, but also in the nature of their vulnerabilities. These findings highlight the importance of developing unlearning techniques that can withstand a diverse range of relearning attacks.

| Relearn | Method | A ↓ | | B ↓ | | C ↓ | |
|---------|--------|-----|--------|-----|--------|-----|--------|
| | | MCQ | Corpus | MCQ | Corpus | MCQ | Corpus |
| A | RMU | — | — | 0.41 | 0.45 | 0.45 | 0.49 |
| A | L-RMU | — | — | **0.40** | 0.49 | 0.50 | 0.54 |
| A | SimNPO | — | — | 0.47 | 0.45 | 0.50 | 0.54 |
| A | L-SimNPO | — | — | 0.41 | **0.35** | **0.41** | **0.40** |
| B | RMU | 0.41 | 0.48 | — | — | 0.46 | 0.48 |
| B | L-RMU | **0.31** | **0.44** | — | — | 0.50 | 0.54 |
| B | SimNPO | 0.54 | 0.52 | — | — | 0.54 | 0.53 |
| B | L-SimNPO | 0.42 | 0.45 | — | — | **0.40** | **0.41** |
| C | RMU | 0.43 | 0.50 | 0.39 | 0.44 | — | — |
| C | L-RMU | **0.26** | **0.36** | **0.34** | 0.42 | — | — |
| C | SimNPO | 0.52 | 0.55 | 0.48 | 0.44 | — | — |
| C | L-SimNPO | 0.42 | 0.47 | 0.45 | **0.42** | — | — |

Table 3: Relearning accuracies on WMDP for RMU, SimNPO, and 3-fold layered variants of both. We see that layered variants are more robust to relearning. This robustness is one-directional for L-RMU and partially bidirectional for L-SimNPO. This also shows that corpus attacks are generally more performance than multiple choice (MCQ) for the RMU variants. Similar results for Years and MMLU are shown in Appendix E.

## 4 Related work

**Unlearning for LLMs.** Machine unlearning for large language models (LLMs) has become an active area of research (Lu et al., 2022; Jang et al., 2022; Kumar et al., 2022; Zhang et al., 2023; Pawelczyk et al., 2023; Eldan & Russinovich, 2023; Ishibashi & Shimodaira, 2023; Yao et al., 2023; Maini et al., 2024; Zhang et al., 2024b; Li et al., 2024b; Wang et al., 2024; Jia et al., 2024; Liu et al., 2024b;a; Thaker et al., 2024; Kadhe et al., 2024; Fan et al., 2025; Zhang et al., 2024a). Due to the difficulty of exact unlearning, most existing methods adopt approximate strategies, including model optimization (Ilharco et al., 2022; Liu et al., 2022; Yao et al., 2023; Eldan & Russinovich, 2023; Jia et al., 2024; Zhang et al., 2024b; Li et al., 2024b) and prompt-based or in-context learning techniques (Thaker et al., 2024; Pawelczyk et al., 2023; Liu et al., 2024a). However, recent work has shown that these models often remain vulnerable to adversarial attacks (Schwarzschild et al., 2024; Patil et al., 2024; Lynch et al., 2024) or to relearning from small fragments of previously seen data (Hu et al., 2024; Lynch et al., 2024). These findings highlight the persistent challenges in achieving robust unlearning in LLMs.

**Adversarial relearning.** Adversarial relearning attacks exploit residual knowledge after unlearning by fine-tuning on a small subset of forgotten data, aiming to recover information about the full unlearned set. Che et al. (2024) showed that most existing unlearning methods are vulnerable to such attacks, revealing a fundamental limitation. Deeb & Roger (2025) further demonstrated that even informationally distinct examples can induce relearning, indicating failures beyond rote memorization. While several defenses have been proposed (Rosati et al., 2024; Zou et al., 2024; Tamirisa et al., 2025; Sheshadri et al., 2025), none have consistently withstood adversarial relearning (Che et al., 2024).

Prior work has studied both corpus-based (Che et al., 2024) and MCQ-based (Deeb & Roger, 2025) fine-tuning. To our knowledge, no comprehensive comparison of the two strategies has been conducted; we find corpus-based fine-tuning to be a more natural and effective form of adversarial relearning.

**Sequential unlearning.** Sequential unlearning has been explored in various contexts, such as removing copyrighted information over time (Dou et al., 2025). Zhao et al. (2024) studied sequential unlearning as a means to improve forgetting efficiency but did not consider its impact on robustness against adversarial relearning. In contrast, our work investigates how the order and structure of sequential unlearning influence robustness, focusing on its potential to mitigate adversarial relearning. Specifically, we analyze the path dependence of unlearning and propose a novel framework that leverages structured forgetting to enhance resilience against information leakage.

## 5 Discussion

LU requires separate hyperparameter tuning at each forgetting stage to balance retention and forgetting. As the number of folds increases, the model must effectively discriminate between each pair of folds, with complexity scaling as $\binom{k}{2}$, which limits scalability to large $k$. Furthermore, LU is by nature more computationally intensive. However, in exchange for taking more time, it discovers optima that standard unlearning techniques are unable to discover no matter how long they train. Future work could consider more efficient methods.

Comprehensive adversarial evaluation is also difficult due to the exponential number of relearning subsets ($2^k - 2$) and additional attack configurations (e.g., batch size, learning rate, dataset, unlearning method). While we leave a full analysis to future work, our fixed hyperparameter setting was sufficient to break all baseline methods (RMU, SimNPO), as detailed in the Appendix.

We do not directly address the challenge of harmless fine-tuning, where the attacker uses data unrelated to what was unlearned, but we offer an intuition to guide future work. In standard unlearning, relearning is significantly easier when the attacker has access to data that is similar to the unlearned examples. Even small amounts of related data can serve as powerful signals, making it surprisingly effective to recover forgotten information. We hypothesize that LU reduces this vulnerability by making recovery difficult even when related data is available. As a result, it shrinks the performance gap between fine-tuning with related versus unrelated data, potentially making both equally ineffective.

Finally, we consider how the structure of LU might inform post-training more broadly. Betley et al. (2025) show that fine-tuning on a single behavior—such as insecure code—can unintentionally induce harmful behaviors, suggesting entanglement between seemingly unrelated capabilities and values. One possible explanation is that a single post-training run introduces a shared inhibitor that influences multiple behaviors at once. LU, by contrast, creates multiple, distinct inhibitors and may help disentangle these behaviors. This perspective suggests a potential alignment strategy: first train a model to be harmless but helpless, then fine-tune it to be helpful while preserving harmlessness. In this setup, our results suggest increasing helplessness should preserve harmlessness, while increasing harmfulness should increase helplessness. While we focus on unlearning, we believe this layered approach could extend to alignment and other post-training interventions, offering a possible path toward more modular and controllable model behavior.

## 6 Conclusion

We introduced **Layered Unlearning**, a $k$-fold sequential unlearning framework that improves robustness by constructing functionally distinct, context-dependent inhibitors. Our experiments demonstrate that LU reliably blocks recovery of earlier folds and significantly improves robustness across both synthetic and LLM benchmarks. While LU strengthens defenses against standard MCQ-based fine-tuning, it also reveals the limitations of current methods when faced with stronger corpus-based attacks. These results suggest that forgetting is inherently brittle and that robustness requires structured, layered defenses. More broadly, LU offers a testbed for mechanistic investigations of inhibitors and a conceptual foundation for more resilient post-training interventions.

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

## A    Layered Unlearning graphics

We provide graphics to better communicate the main idea.

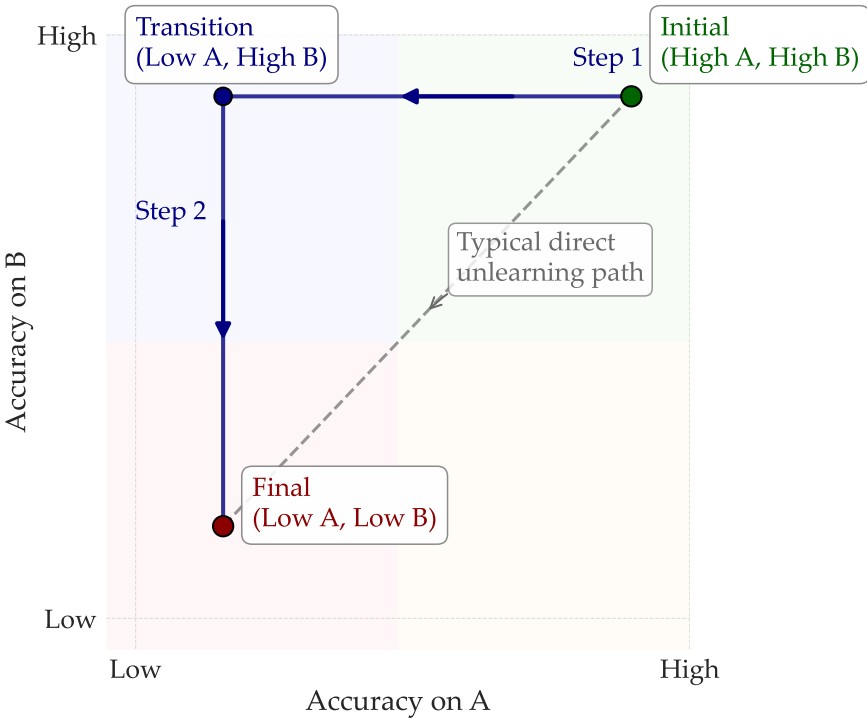

Figure 4: The performance trajectory of LU on two folds $A, B$. Normally, unlearning methods lose performance on $A, B$ jointly and directly head towards the red point. However, we propose performing LU to retain performance on $B$ while forgetting $A$ and then forgetting both folds.

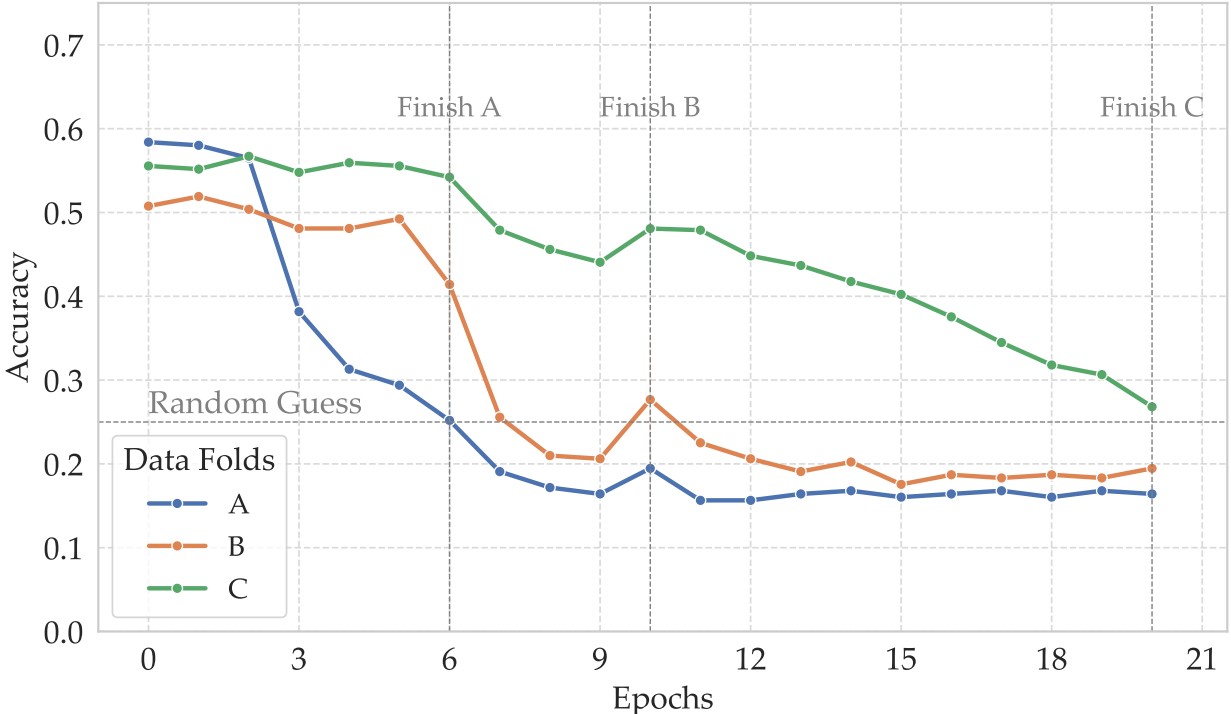

Figure 5: We show the accuracy progression of forgetting three sets $A, B, C$ in that order using RMU on WMDP. The vertical dotted gray lines show when we move to forgetting the next fold. Note that the $A$ accuracy drops in the first iteration of forgetting and remains low. The accuracy of $B$ remains high until the second iteration of forgetting, and then drops and remains low. Finally, the accuracy of $C$ remains high until the third half of forgetting, when it drops.

## B  Synthetic ablation experiments

### B.1  Logistic regression

We investigate different clustering schemes for grouping Gaussians into tasks $A$, $B$, and $R$. In the K-Means setup, we first cluster the Gaussian means using K-Means, then solve a linear assignment problem to evenly assign clusters to tasks based on proximity. Appendix Figures 6 and 7 show that adversarial relearning becomes more effective as the number of clusters increases. In contrast, LU consistently resists relearning, though its robustness is somewhat reduced under random clustering.

Our intuition is that adversarial relearning is less effective when task boundaries are more distinct. K-Means clustering tends to separate tasks more cleanly, thereby limiting overlap. In contrast, random clustering—especially with a larger number of Gaussians—increases the likelihood of overlap between tasks, making it more difficult to defend against adversarial relearning. Additionally, increasing the number of clusters inherently raises the potential for such overlap.

### B.2  Bigram modeling

We analyze the effect of substituting components from the LU model into the U model on adversarial relearning, using the notation of Elhage et al. (2021). Substitutions are grouped as follows:

- $QK$: Replace $W_Q$ and $W_K$.
- $OV$: Replace $W_O$ and $W_V$.

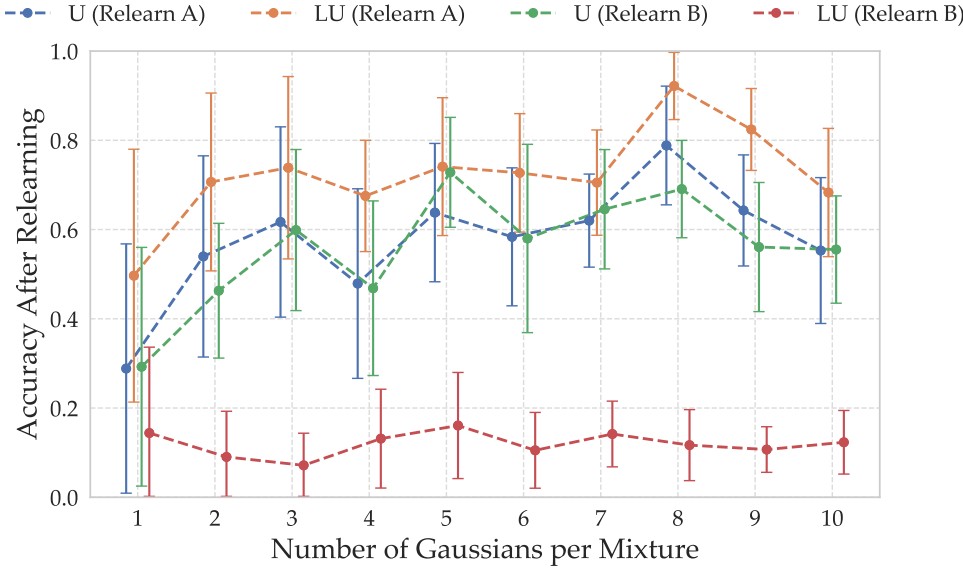

Figure 6: Relearning accuracies on $A$ and $B$ using datasets generated via K-Means clustering. Error bars denote 2-std confidence intervals across 10 random seeds.

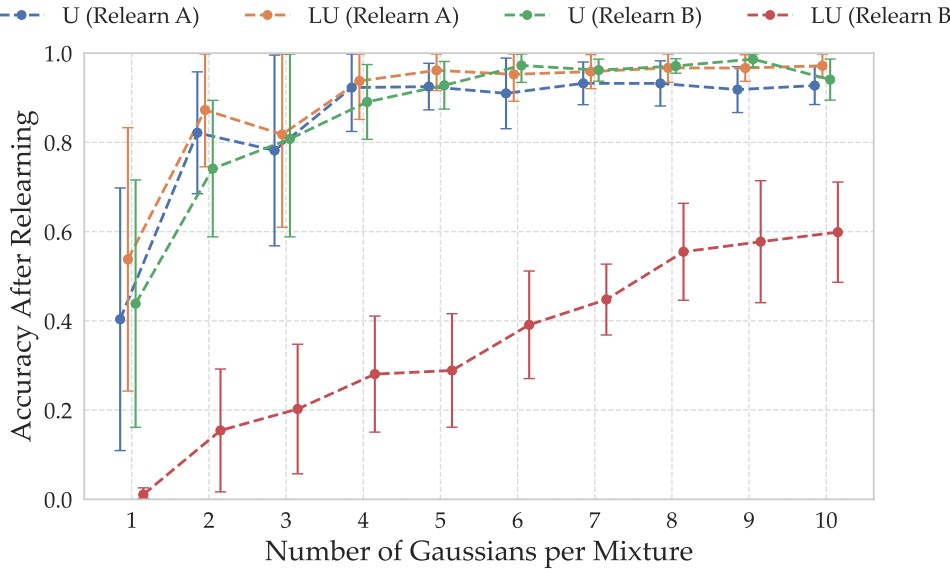

Figure 7: Relearning accuracies on $A$ and $B$ using datasets generated via random clustering. Error bars denote 2-std confidence intervals across 10 random seeds.

- $UE$: Replace $W_U$ and $W_E$.

These groupings reflect functional units in the model. As shown in Appendix Table 4, substituting $QK$ or $OV$ consistently yields the greatest robustness to adversarial relearning, highlighting the key role of attention for this setting. This pattern is also visible in Appendix Figure 8. Notably, retain and task accuracies remain stable across all substitution settings (Appendix Table 5), indicating no degradation in core performance. Investigating how attention confers this robustness is left for future work.

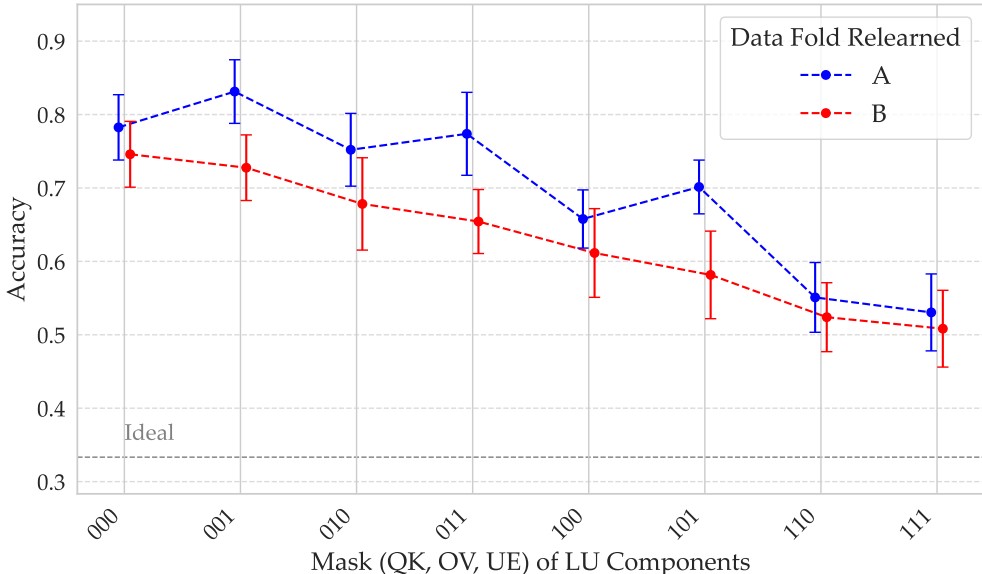

Figure 8: Relearning accuracies on $A$ and $B$ as a function of which components ($QK$, $OV$, $UE$) of the LU model are substituted in. The $x$-axis encodes binary masks (e.g., 000 is standard unlearning; 111 is full LU). Error bars denote 2-std confidence intervals over 10 random seeds. One seed was excluded due to universal resistance to relearning, which only increased error bar size without affecting trends. "Ideal" denotes perfect unlearning.

| Relearn | QK | OV | UE | A ↓ | B ↓ | Retain ↓ |
|---------|----|----|----|------|------|----------|
| A | 0 | 0 | 0 | — | 0.78 | 0.06 |
| A | 0 | 0 | 1 | — | 0.83 | 0.05 |
| A | 0 | 1 | 0 | — | 0.75 | 0.05 |
| A | 0 | 1 | 1 | — | 0.77 | 0.05 |
| A | 1 | 0 | 0 | — | 0.66 | 0.04 |
| A | 1 | 0 | 1 | — | 0.70 | **0.03** |
| A | 1 | 1 | 0 | — | 0.55 | **0.03** |
| A | 1 | 1 | 1 | — | **0.53** | 0.04 |
| B | 0 | 0 | 0 | 0.75 | — | 0.05 |
| B | 0 | 0 | 1 | 0.73 | — | 0.05 |
| B | 0 | 1 | 0 | 0.68 | — | 0.05 |
| B | 0 | 1 | 1 | 0.65 | — | **0.04** |
| B | 1 | 0 | 0 | 0.61 | — | 0.05 |
| B | 1 | 0 | 1 | 0.58 | — | **0.04** |
| B | 1 | 1 | 0 | 0.52 | — | **0.04** |
| B | 1 | 1 | 1 | **0.51** | — | **0.04** |

Table 4: Table of relearning accuracies when substituting different model components. A value of 0 indicates components from the U model, while 1 indicates components from the LU model. The "Retain" column reports total variation (TV) distance on the retain set. This is an average over 10 random seeds.

| QK | OV | UE | A ↓ | B ↓ | Retain ↓ |
|----|----|----|-----|-----|----------|
| 0 | 0 | 0 | 0.33 | 0.33 | 0.01 |
| 0 | 0 | 1 | 0.36 | 0.39 | 0.02 |
| 0 | 1 | 0 | **0.31** | **0.30** | 0.02 |
| 0 | 1 | 1 | 0.32 | 0.34 | 0.02 |
| 1 | 0 | 0 | 0.33 | 0.33 | **0.01** |
| 1 | 0 | 1 | 0.36 | 0.39 | 0.02 |
| 1 | 1 | 0 | 0.32 | **0.30** | 0.02 |
| 1 | 1 | 1 | 0.34 | 0.33 | **0.02** |

Table 5: Unlearned accuracies after substituting model components. A value of 0 indicates components from the U model, and 1 indicates components from the LU model. The "Retain" column reports the total variation (TV) distance on the retain set. This is an average over 10 random seeds.

## C  Retain accuracies

For evaluation, we either use full MMLU evaluation or we use specific categories for MMLU created in Deeb & Roger (2025). The retain categories for MMLU consist of questions relating to health, history, law, philosophy, and the social sciences. The forget categories for MMLU consist of questions relating to geography, culture, STEM, chemistry, and business.

| Size | Dataset | Method | Retain ↑ |
|---|---|---|---|
| Small | — | None | 0.59 |
| Full | — | None | 0.58 |
| Full | WMDP 2 | RMU | 0.57 |
| Full | WMDP 2 | RMU-Split | 0.58 |
| Full | WMDP 2 | L-RMU | 0.57 |
| Full | WMDP 2 | L-RMU-Split | 0.57 |
| Full | WMDP 3 | RMU | 0.57 |
| Full | WMDP 3 | RMU-Split | 0.57 |
| Full | WMDP 3 | L-RMU | 0.57 |
| Full | WMDP 3 | L-RMU-Split | 0.55 |
| Full | WMDP 3 | SimNPO | 0.57 |
| Full | WMDP 3 | L-SimNPO | 0.55 |
| Full | WMDP 4 | RMU | 0.57 |
| Full | WMDP 4 | RMU-Split | 0.57 |
| Full | WMDP 4 | L-RMU | 0.57 |
| Full | WMDP 4 | L-RMU-Split | 0.57 |
| Small | MMLU 3 | RMU | 0.55 |
| Small | MMLU 3 | RMU-Split | 0.56 |
| Small | MMLU 3 | L-RMU | 0.54 |
| Small | MMLU 3 | L-RMU-Split | 0.55 |
| Full | Years 3 | RMU | 0.58 |
| Full | Years 3 | RMU-Split | 0.58 |
| Full | Years 3 | L-RMU | 0.58 |
| Full | Years 3 | L-RMU-Split | 0.57 |

Table 6: Retain accuracy by method. We refer to the restricted subset of MMLU questions as the *small set* and the full MMLU dataset as the *full set*. The dataset column indicates which dataset was unlearned for each method. The first section shows results from the original model before any unlearning was applied.

## D  Unlearning accuracies

We provide the accuracies after applying the unlearning methods on each fold for each dataset. We also analyze another variant of RMU, which we term RMU-Split. In RMU-Split, each fold is projected onto a different random vector, in contrast to the shared projection used in RMU. This isolates the impact of LU from the confounding effect of using separate random vectors. The results do not change.

| Dataset | Method | A ↓ | B ↓ | C ↓ | D ↓ |
|---------|--------|-----|-----|-----|-----|
| WMDP 2 | RMU | 0.26 | 0.29 | — | — |
| WMDP 2 | L-RMU | **0.21** | 0.28 | — | — |
| WMDP 2 | RMU-Split | 0.25 | **0.26** | — | — |
| WMDP 2 | L-RMU-Split | 0.24 | 0.28 | — | — |
| WMDP 3 | RMU | 0.27 | 0.24 | 0.33 | — |
| WMDP 3 | L-RMU | **0.15** | 0.24 | 0.33 | — |
| WMDP 3 | RMU-Split | 0.22 | **0.18** | **0.22** | — |
| WMDP 3 | L-RMU-Split | 0.16 | 0.19 | 0.27 | — |
| WMDP 3 | SimNPO | 0.32 | 0.24 | 0.31 | — |
| WMDP 3 | L-SimNPO | 0.29 | 0.32 | 0.33 | — |
| WMDP 4 | RMU | 0.25 | 0.27 | 0.29 | 0.30 |
| WMDP 4 | L-RMU | **0.15** | **0.21** | 0.25 | 0.34 |
| WMDP 4 | RMU-Split | 0.28 | 0.26 | **0.23** | **0.22** |
| WMDP 4 | L-RMU-Split | 0.18 | 0.23 | 0.27 | 0.34 |
| MMLU 3 | RMU | 0.28 | 0.30 | **0.30** | — |
| MMLU 3 | L-RMU | 0.23 | 0.29 | 0.34 | — |
| MMLU 3 | RMU-Split | 0.26 | 0.34 | 0.30 | — |
| MMLU 3 | L-RMU-Split | **0.21** | **0.29** | 0.32 | — |
| Years 3 | RMU | 0.30 | 0.22 | 0.30 | — |
| Years 3 | L-RMU | **0.29** | 0.28 | 0.31 | — |
| Years 3 | RMU-Split | 0.33 | 0.29 | **0.22** | — |
| Years 3 | L-RMU-Split | 0.29 | **0.20** | 0.27 | — |

Table 7: Unlearning accuracy by method across datasets and evaluation folds.

# E  Relearning accuracies

All attacks are performed with learning rate $10^{-6}$ and batch size 4. We fine-tune on MCQ for 8 epochs and fine-tune on corpus for 5 epochs. This difference is because we wish to fine-tune until the accuracy on the relearn set stops increasing for a while, and by definition fine-tuning on MCQ can reach 1.0 accuracy, so we fine-tune for longer. We then take the maximum validation accuracy across all epochs.

## E.1  WMDP 2 folds

Table 8: Relearning accuracy across methods for WMDP 2 folds.

| Relearn | Method | A ↓ | | B ↓ | |
|---------|--------|-----|--------|-----|--------|
| | | MCQ | Corpus | MCQ | Corpus |
| A | RMU | — | — | 0.45 | **0.48** |
| A | L-RMU | — | — | 0.43 | 0.53 |
| A | RMU-Split | — | — | **0.42** | 0.51 |
| A | L-RMU-Split | — | — | 0.42 | 0.54 |
| B | RMU | 0.45 | 0.51 | — | — |
| B | L-RMU | **0.30** | **0.41** | — | — |
| B | RMU-Split | 0.40 | 0.53 | — | — |
| B | L-RMU-Split | 0.32 | 0.48 | — | — |

### E.2 WMDP 3 folds

Table 9: Relearning accuracy across methods for WMDP 3 folds.

| Relearn | Method | A ↓ | | B ↓ | | C ↓ | |
|---------|--------|-----|--------|-----|--------|-----|--------|
| | | MCQ | Corpus | MCQ | Corpus | MCQ | Corpus |
| A | RMU | — | — | 0.41 | 0.45 | 0.45 | 0.49 |
| A | L-RMU | — | — | 0.40 | 0.49 | 0.50 | 0.54 |
| A | RMU-Split | — | — | 0.37 | 0.44 | **0.38** | 0.54 |
| A | L-RMU-Split | — | — | **0.35** | 0.45 | 0.49 | 0.55 |
| A | SimNPO | — | — | 0.47 | 0.45 | 0.50 | 0.54 |
| A | L-SimNPO | — | — | 0.41 | **0.35** | 0.41 | **0.40** |
| B | RMU | 0.41 | 0.48 | — | — | 0.46 | 0.48 |
| B | L-RMU | 0.31 | 0.44 | — | — | 0.50 | 0.54 |
| B | RMU-Split | 0.42 | 0.53 | — | — | **0.38** | 0.54 |
| B | L-RMU-Split | **0.29** | **0.43** | — | — | 0.47 | 0.54 |
| B | SimNPO | 0.54 | 0.52 | — | — | 0.54 | 0.53 |
| B | L-SimNPO | 0.42 | 0.45 | — | — | 0.40 | **0.41** |
| C | RMU | 0.43 | 0.50 | 0.39 | 0.44 | — | — |
| C | L-RMU | 0.26 | 0.36 | 0.34 | 0.42 | — | — |
| C | RMU-Split | 0.40 | 0.55 | 0.39 | 0.45 | — | — |
| C | L-RMU-Split | **0.25** | **0.29** | **0.27** | **0.34** | — | — |
| C | SimNPO | 0.52 | 0.55 | 0.48 | 0.44 | — | — |
| C | L-SimNPO | 0.42 | 0.47 | 0.45 | 0.42 | — | — |
| A, B | RMU | — | — | — | — | 0.48 | 0.51 |
| A, B | L-RMU | — | — | — | — | 0.49 | 0.56 |
| A, B | RMU-Split | — | — | — | — | **0.41** | 0.55 |
| A, B | L-RMU-Split | — | — | — | — | 0.50 | 0.54 |
| A, B | SimNPO | — | — | — | — | 0.57 | 0.53 |
| A, B | L-SimNPO | — | — | — | — | 0.43 | **0.40** |
| A, C | RMU | — | — | 0.43 | 0.47 | — | — |
| A, C | L-RMU | — | — | 0.38 | 0.50 | — | — |
| A, C | RMU-Split | — | — | 0.42 | 0.45 | — | — |
| A, C | L-RMU-Split | — | — | **0.34** | 0.45 | — | — |
| A, C | SimNPO | — | — | 0.48 | 0.44 | — | — |
| A, C | L-SimNPO | — | — | 0.44 | **0.40** | — | — |
| B, C | RMU | 0.43 | 0.54 | — | — | — | — |
| B, C | L-RMU | **0.34** | 0.47 | — | — | — | — |
| B, C | RMU-Split | 0.39 | 0.56 | — | — | — | — |
| B, C | L-RMU-Split | 0.35 | **0.42** | — | — | — | — |
| B, C | SimNPO | 0.52 | 0.56 | — | — | — | — |
| B, C | L-SimNPO | 0.44 | 0.47 | — | — | — | — |

### E.3 WMDP 4 folds

Table 10: Relearning accuracy across methods for WMDP 4 folds.

| Relearn | Method | A ↓ | | B ↓ | | C ↓ | | D ↓ | |
|---|---|---|---|---|---|---|---|---|---|
| | | MCQ | Corpus | MCQ | Corpus | MCQ | Corpus | MCQ | Corpus |
| A | RMU | — | — | 0.41 | **0.44** | 0.37 | **0.44** | 0.41 | **0.47** |
| A | L-RMU | — | — | 0.38 | 0.46 | **0.35** | 0.46 | **0.40** | 0.57 |
| A | RMU-Split | — | — | 0.42 | 0.46 | 0.38 | 0.44 | 0.41 | 0.51 |
| A | L-RMU-Split | — | — | **0.38** | 0.50 | 0.39 | 0.47 | 0.44 | 0.56 |
| B | RMU | 0.46 | **0.47** | — | — | 0.44 | **0.41** | 0.49 | **0.44** |
| B | L-RMU | **0.29** | 0.51 | — | — | **0.36** | 0.46 | 0.46 | 0.56 |
| B | RMU-Split | 0.42 | 0.51 | — | — | 0.37 | 0.45 | **0.40** | 0.51 |
| B | L-RMU-Split | 0.33 | **0.47** | — | — | 0.37 | 0.47 | 0.51 | 0.56 |
| C | RMU | 0.44 | 0.49 | 0.43 | 0.47 | — | — | 0.46 | **0.45** |
| C | L-RMU | 0.33 | **0.45** | 0.37 | 0.47 | — | — | 0.46 | 0.57 |
| C | RMU-Split | 0.43 | 0.51 | **0.35** | **0.46** | — | — | **0.41** | 0.49 |
| C | L-RMU-Split | **0.29** | 0.47 | **0.35** | 0.47 | — | — | 0.45 | 0.55 |
| D | RMU | 0.43 | 0.47 | 0.42 | 0.47 | 0.40 | 0.42 | — | — |
| D | L-RMU | **0.25** | **0.33** | **0.27** | **0.42** | 0.36 | 0.47 | — | — |
| D | RMU-Split | 0.37 | 0.52 | 0.38 | 0.45 | 0.34 | 0.45 | — | — |
| D | L-RMU-Split | **0.25** | 0.40 | 0.32 | 0.43 | **0.32** | **0.40** | — | — |
| A, B | RMU | — | — | — | — | 0.41 | 0.47 | 0.48 | **0.53** |
| A, B | L-RMU | — | — | — | — | 0.43 | 0.49 | 0.46 | 0.56 |
| A, B | RMU-Split | — | — | — | — | **0.38** | **0.45** | 0.43 | **0.53** |
| A, B | L-RMU-Split | — | — | — | — | 0.39 | 0.48 | 0.57 | 0.56 |
| A, C | RMU | — | — | 0.41 | 0.49 | — | — | 0.50 | **0.49** |
| A, C | L-RMU | — | — | 0.43 | 0.48 | — | — | 0.46 | 0.57 |
| A, C | RMU-Split | — | — | **0.38** | 0.48 | — | — | **0.39** | 0.53 |
| A, C | L-RMU-Split | — | — | 0.41 | 0.52 | — | — | 0.48 | 0.56 |
| A, D | RMU | — | — | 0.43 | 0.49 | 0.40 | **0.45** | — | — |
| A, D | L-RMU | — | — | **0.35** | 0.48 | 0.38 | 0.46 | — | — |
| A, D | RMU-Split | — | — | 0.46 | 0.49 | 0.41 | 0.47 | — | — |
| A, D | L-RMU-Split | — | — | 0.44 | 0.51 | 0.40 | 0.48 | — | — |
| B, C | RMU | 0.47 | 0.53 | — | — | — | — | 0.48 | **0.51** |
| B, C | L-RMU | **0.30** | 0.53 | — | — | — | — | 0.46 | 0.56 |
| B, C | RMU-Split | 0.43 | 0.55 | — | — | — | — | **0.40** | 0.53 |
| B, C | L-RMU-Split | 0.35 | **0.49** | — | — | — | — | 0.56 | 0.56 |
| B, D | RMU | 0.45 | 0.56 | — | — | 0.44 | **0.46** | — | — |
| B, D | L-RMU | 0.35 | 0.51 | — | — | 0.42 | 0.47 | — | — |
| B, D | RMU-Split | 0.45 | 0.55 | — | — | **0.39** | **0.46** | — | — |
| B, D | L-RMU-Split | **0.34** | **0.49** | — | — | 0.39 | 0.49 | — | — |
| C, D | RMU | 0.43 | 0.54 | 0.42 | 0.48 | — | — | — | — |
| C, D | L-RMU | **0.27** | **0.47** | **0.35** | **0.45** | — | — | — | — |
| C, D | RMU-Split | 0.36 | 0.55 | 0.38 | 0.48 | — | — | — | — |
| C, D | L-RMU-Split | 0.29 | 0.48 | 0.41 | 0.51 | — | — | — | — |

*Continued on next page*

*Continued from previous page*

| Relearn | Method | A ↓ | | B ↓ | | C ↓ | | D ↓ | |
|---------|--------|-----|--------|-----|--------|-----|--------|-----|--------|
| | | MCQ | Corpus | MCQ | Corpus | MCQ | Corpus | MCQ | Corpus |
| A, B, C | RMU | — | — | — | — | — | — | 0.46 | **0.52** |
| A, B, C | L-RMU | — | — | — | — | — | — | 0.49 | 0.56 |
| A, B, C | RMU-Split | — | — | — | — | — | — | **0.42** | 0.53 |
| A, B, C | L-RMU-Split | — | — | — | — | — | — | 0.59 | 0.56 |
| A, B, D | RMU | — | — | — | — | 0.44 | 0.48 | — | — |
| A, B, D | L-RMU | — | — | — | — | **0.42** | 0.51 | — | — |
| A, B, D | RMU-Split | — | — | — | — | **0.42** | **0.47** | — | — |
| A, B, D | L-RMU-Split | — | — | — | — | 0.45 | 0.49 | — | — |
| A, C, D | RMU | — | — | 0.48 | 0.49 | — | — | — | — |
| A, C, D | L-RMU | — | — | **0.39** | **0.48** | — | — | — | — |
| A, C, D | RMU-Split | — | — | 0.49 | 0.49 | — | — | — | — |
| A, C, D | L-RMU-Split | — | — | 0.43 | 0.51 | — | — | — | — |
| B, C, D | RMU | 0.47 | 0.55 | — | — | — | — | — | — |
| B, C, D | L-RMU | **0.31** | 0.51 | — | — | — | — | — | — |
| B, C, D | RMU-Split | 0.44 | 0.57 | — | — | — | — | — | — |
| B, C, D | L-RMU-Split | 0.37 | **0.48** | — | — | — | — | — | — |

### E.4 MMLU 3 folds

Table 11: Relearning accuracy across methods for MMLU 3 folds.

| Relearn | Method | A ↓ | | B ↓ | | C ↓ | |
|---|---|---|---|---|---|---|---|
| | | MCQ | Corpus | MCQ | Corpus | MCQ | Corpus |
| A | RMU | — | — | 0.49 | 0.66 | 0.51 | 0.64 |
| A | L-RMU | — | — | **0.40** | 0.61 | 0.56 | 0.64 |
| A | RMU-Split | — | — | 0.52 | 0.63 | **0.48** | **0.62** |
| A | L-RMU-Split | — | — | **0.40** | **0.57** | 0.50 | 0.64 |
| B | RMU | 0.52 | 0.65 | — | — | 0.51 | **0.62** |
| B | L-RMU | **0.30** | **0.54** | — | — | 0.59 | 0.64 |
| B | RMU-Split | 0.50 | 0.63 | — | — | **0.49** | 0.62 |
| B | L-RMU-Split | 0.38 | 0.61 | — | — | **0.49** | 0.63 |
| C | RMU | 0.53 | 0.64 | 0.52 | 0.64 | — | — |
| C | L-RMU | **0.33** | **0.39** | 0.39 | **0.46** | — | — |
| C | RMU-Split | 0.55 | 0.65 | 0.57 | 0.64 | — | — |
| C | L-RMU-Split | 0.36 | 0.48 | **0.38** | 0.50 | — | — |
| A, B | RMU | — | — | — | — | 0.54 | 0.63 |
| A, B | L-RMU | — | — | — | — | 0.60 | 0.64 |
| A, B | RMU-Split | — | — | — | — | 0.61 | 0.64 |
| A, B | L-RMU-Split | — | — | — | — | **0.53** | **0.63** |
| A, C | RMU | — | — | 0.55 | 0.64 | — | — |
| A, C | L-RMU | — | — | **0.44** | 0.61 | — | — |
| A, C | RMU-Split | — | — | 0.61 | 0.66 | — | — |
| A, C | L-RMU-Split | — | — | 0.48 | **0.59** | — | — |
| B, C | RMU | 0.52 | 0.65 | — | — | — | — |
| B, C | L-RMU | **0.32** | **0.55** | — | — | — | — |
| B, C | RMU-Split | 0.53 | 0.67 | — | — | — | — |
| B, C | L-RMU-Split | 0.38 | 0.60 | — | — | — | — |

### E.5 Years 3 folds

Table 12: Relearning accuracy across methods for Years 3 folds.

| Relearn | Method | A ↓ | | B ↓ | | C ↓ | |
|---|---|---|---|---|---|---|---|
| | | MCQ | Corpus | MCQ | Corpus | MCQ | Corpus |
| A | RMU | — | — | 0.55 | 0.57 | 0.55 | 0.58 |
| A | L-RMU | — | — | 0.47 | 0.45 | **0.50** | 0.53 |
| A | RMU-Split | — | — | 0.55 | 0.48 | 0.54 | **0.48** |
| A | L-RMU-Split | — | — | **0.43** | **0.36** | 0.51 | 0.50 |
| B | RMU | 0.56 | 0.59 | — | — | 0.55 | 0.58 |
| B | L-RMU | 0.40 | 0.33 | — | — | 0.51 | **0.51** |
| B | RMU-Split | 0.59 | 0.48 | — | — | **0.49** | 0.52 |
| B | L-RMU-Split | **0.37** | **0.33** | — | — | 0.52 | 0.52 |
| C | RMU | 0.54 | 0.58 | 0.58 | 0.58 | — | — |
| C | L-RMU | **0.29** | 0.32 | 0.43 | 0.42 | — | — |
| C | RMU-Split | 0.54 | 0.46 | 0.54 | 0.47 | — | — |
| C | L-RMU-Split | 0.29 | **0.31** | **0.42** | **0.38** | — | — |
| A, B | RMU | — | — | — | — | 0.61 | 0.58 |
| A, B | L-RMU | — | — | — | — | **0.51** | **0.52** |
| A, B | RMU-Split | — | — | — | — | 0.59 | 0.54 |
| A, B | L-RMU-Split | — | — | — | — | 0.56 | 0.53 |
| A, C | RMU | — | — | 0.61 | 0.57 | — | — |
| A, C | L-RMU | — | — | 0.52 | 0.45 | — | — |
| A, C | RMU-Split | — | — | 0.60 | 0.51 | — | — |
| A, C | L-RMU-Split | — | — | **0.51** | **0.42** | — | — |
| B, C | RMU | 0.62 | 0.59 | — | — | — | — |
| B, C | L-RMU | 0.43 | **0.32** | — | — | — | — |
| B, C | RMU-Split | 0.59 | 0.50 | — | — | — | — |
| B, C | L-RMU-Split | **0.40** | 0.34 | — | — | — | — |

# F   Hyperparameters

We set the forgetting threshold to 0.35, motivated by the following statistical reasoning.

For multiple-choice questions (MCQ) with 4 choices, assuming random guessing, the expected accuracy is 0.25. Each dataset contains a total of 735 questions, and the smallest fold we consider consists of $\frac{735}{4}$. By applying the central limit theorem, the expected final accuracy follows approximately a normal distribution:

$$\mathcal{N}\left(\frac{1}{4}, \sqrt{\frac{1}{4} \cdot \frac{3}{4} \cdot \frac{4}{735}}\right) \approx \mathcal{N}\left(0.25, 0.032\right).$$

A three-standard-deviation event corresponds to:

$$3 \times 0.032 + 0.25 \leq 0.35.$$

Since we do not reject the null hypothesis of random guessing if accuracy remains within three standard deviations of 0.25, we set the forgetting threshold to approximately 0.35. Note that this bound does get tighter if we consider bigger folds, but in practice, we find that this threshold does not significantly impact results, so we standardize it across all instances of LU.

For all models, we use Zephyr-7$B$-$\beta$ (Tunstall et al., 2023).

## F.1   RMU hyperparameters

WMDP: official model checkpoint from (Li et al., 2024a). MMLU:

- Activation layer: 7.
- Layers fine-tuned: 5, 6, 7.
- Magnitude: 10.
- Forget coefficient: 2.00.
- Retain coefficient: 16.00.
- Learning rate: $5 \times 10^{-5}$.
- Batch size: 4.

Years:

- Activation layer: 7.
- Layers fine-tuned: 5, 6, 7.
- Magnitude: 15.
- Forget coefficient: 0.25.
- Retain coefficient: 1.00.
- Learning rate: $5 \times 10^{-5}$.
- Batch size: 4.

### F.2   RMU-Split hyperparameters

WMDP 2 folds:

- Activation layer: 7.
- Layers fine-tuned: $5, 6, 7$.
- Magnitude: 10.
- Forget coefficients: $1.00, 1.00$.
- Retain coefficient 32.
- Learning rate: $1 \times 10^{-5}$.
- Batch size: 4.

WMDP 3 folds:

- Activation layer: 7.
- Layers fine-tuned: $5, 6, 7$.
- Magnitude: 10.
- Forget coefficients: $1.00, 1.00, 1.00$.
- Retain coefficient 16.
- Learning rate: $1 \times 10^{-5}$.
- Batch size: 4.

WMDP 4 folds:

- Activation layer: 7.
- Layers fine-tuned: $5, 6, 7$.
- Magnitude: 10.
- Forget coefficients: $1.00, 1.00, 1.00, 1.00$.
- Retain coefficient 32.
- Learning rate: $1 \times 10^{-5}$.
- Batch size: 4.

MMLU 3 folds:

- Activation layer: 7.
- Layers fine-tuned: $5, 6, 7$.
- Magnitude: 10.
- Forget coefficients: $2.00, 2.00, 2.00$.
- Retain coefficient 24.

- MMLU retain coefficient: 12.0.

- Learning rate: $1 \times 10^{-5}$.

- Batch size: 8.

Years 3 folds:

- Activation layer: 7.

- Layers fine-tuned: $5, 6, 7$.

- Magnitude: 10.

- Forget coefficients: $1.00, 1.00, 1.00$.

- Retain coefficient 32.

- Learning rate: $1 \times 10^{-5}$.

- Batch size: 4.

### F.3 L-RMU hyperparameters

WMDP 2 folds:

- Stage 1:
  - Activation layer: 7.
  - Layers fine-tuned: $5, 6, 7$.
  - Magnitude: 6.5.
  - Forget coefficients: $0.39, 0.00$.
  - Retain coefficients: $0.00, 13.52$.
  - Retain set coefficient: 1.
  - Learning rate: $1 \times 10^{-5}$.
  - Batch size: 4.

- Stage 2:
  - Activation layer: 7.
  - Layers fine-tuned: $5, 6, 7$.
  - Magnitude: 6.5.
  - Forget coefficients: $0.10, 1.00$.
  - Retain coefficients: $0.00, 0.00$.
  - Retain set coefficient: 24.
  - Learning rate: $1 \times 10^{-5}$.
  - Batch size: 8.

WMDP 3 folds:

- Stage 1:
  - Activation layer: 7.
  - Layers fine-tuned: $5, 6, 7$.
  - Magnitude: 6.5.
  - Forget coefficients: $0.39, 0.00, 0.00$.

- Retain coefficients: $0.00, 6.76, 6.76$.
- Retain set coefficient: $1$.
- Learning rate: $1 \times 10^{-5}$.
- Batch size: $4$.

- Stage 2:

    - Activation layer: $7$.
    - Layers fine-tuned: $5, 6, 7$.
    - Magnitude: $6.5$.
    - Forget coefficients: $0.05, 0.50, 0.00$.
    - Retain coefficients: $0.00, 0.00, 8.00$.
    - Retain set coefficient: $16$.
    - Learning rate: $1 \times 10^{-5}$.
    - Batch size: $8$.

- Stage 3:

    - Activation layer: $7$.
    - Layers fine-tuned: $5, 6, 7$.
    - Magnitude: $6.5$.
    - Forget coefficients: $0.00, 0.03, 0.33$.
    - Retain coefficients: $0.00, 0.00, 0.00$.
    - Retain set coefficient: $24$.
    - Learning rate: $1 \times 10^{-5}$.
    - Batch size: $8$.

WMDP 4 folds:

- Stage 1:

    - Activation layer: $7$.
    - Layers fine-tuned: $5, 6, 7$.
    - Magnitude: $6.5$.
    - Forget coefficients: $0.39, 0.00, 0.00, 0.00$.
    - Retain coefficients: $0.00, 10.67, 10.67, 10.67$.
    - Retain set coefficient: $1$.
    - Learning rate: $1 \times 10^{-5}$.
    - Batch size: $4$.

- Stage 2:

    - Activation layer: $7$.
    - Layers fine-tuned: $5, 6, 7$.
    - Magnitude: $6.5$.
    - Forget coefficients: $0.10, 1.00, 0.00, 0.00$.
    - Retain coefficients: $0.00, 0.00, 4.00, 4.00$.
    - Retain set coefficient: $16$.
    - Learning rate: $1 \times 10^{-5}$.
    - Batch size: $8$.

- Stage 3:

- – Activation layer: 7.
- – Layers fine-tuned: $5, 6, 7$.
- – Magnitude: 6.5.
- – Forget coefficients: $0.01, 0.07, 0.67, 0.00$.
- – Retain coefficients: $0.00, 0.00, 0.00, 8.00$.
- – Retain set coefficient: 16.
- – Learning rate: $1 \times 10^{-5}$.
- – Batch size: 8.

- Stage 4:
  - – Activation layer: 7.
  - – Layers fine-tuned: $5, 6, 7$.
  - – Magnitude: 6.5.
  - – Forget coefficients: $0.01, 0.03, 0.15, 0.75$.
  - – Retain coefficients: $0.00, 0.00, 0.00, 0.00$.
  - – Retain set coefficient: 32.
  - – Learning rate: $1 \times 10^{-5}$.
  - – Batch size: 8.

MMLU 3 folds:

- Stage 1:
  - – Activation layer: 7.
  - – Layers fine-tuned: $5, 6, 7$.
  - – Magnitude: 6.5.
  - – Forget coefficients: $2.00, 0.00, 0.00$.
  - – Retain coefficients: $0.00, 8.00, 8.00$.
  - – Retain set coefficient: 2.
  - – Learning rate: $1 \times 10^{-5}$.
  - – Batch size: 4.

- Stage 2:
  - – Activation layer: 7.
  - – Layers fine-tuned: $5, 6, 7$.
  - – Magnitude: 6.5.
  - – Forget coefficients: $0.10, 2.00, 0.00$.
  - – Retain coefficients: $0.00, 0.00, 4.00$.
  - – Retain set coefficient: 32.
  - – MMLU retain coefficient: 8.0.
  - – Learning rate: $1 \times 10^{-5}$.
  - – Batch size: 8.

- Stage 3:
  - – Activation layer: 7.
  - – Layers fine-tuned: $5, 6, 7$.
  - – Magnitude: 6.5.
  - – Forget coefficients: $0.01, 0.13, 2.67$.
  - – Retain coefficients: $0.00, 0.00, 0.00$.

- Retain set coefficient: 36.
- MMLU retain coefficient: 18.0.
- Learning rate: $1 \times 10^{-5}$.
- Batch size: 8.

Years 3 folds:

- Stage 1:
  - Activation layer: 7.
  - Layers fine-tuned: $5, 6, 7$.
  - Magnitude: 6.5.
  - Forget coefficients: $4.00, 0.00, 0.00$.
  - Retain coefficients: $0.00, 16.00, 16.00$.
  - Retain set coefficient: 2.
  - Learning rate: $1 \times 10^{-5}$.
  - Batch size: 4.

- Stage 2:
  - Activation layer: 7.
  - Layers fine-tuned: $5, 6, 7$.
  - Magnitude: 6.5.
  - Forget coefficients: $2.25, 22.50, 0.00$.
  - Retain coefficients: $0.00, 0.00, 16.00$.
  - Retain set coefficient: 16.
  - Learning rate: $1 \times 10^{-5}$.
  - Batch size: 8.

- Stage 3:
  - Activation layer: 7.
  - Layers fine-tuned: $5, 6, 7$.
  - Magnitude: 6.5.
  - Forget coefficients: $0.15, 1.50, 15.00$.
  - Retain coefficients: $0.00, 0.00, 0.00$.
  - Retain set coefficient: 32.
  - Learning rate: $1 \times 10^{-5}$.
  - Batch size: 8.

### F.4 L-RMU-Split hyperparameters

WMDP 2 folds:

- Stage 1:
  - Activation layer: 7.
  - Layers fine-tuned: $5, 6, 7$.
  - Magnitude: 6.5.
  - Forget coefficients: $0.39, 0.00$.
  - Retain coefficients: $0.00, 13.52$.
  - Retain set coefficient: 1.

- Learning rate: $1 \times 10^{-5}$.
- Batch size: 4.

- Stage 2:

    - Activation layer: 7.
    - Layers fine-tuned: $5, 6, 7$.
    - Magnitude: 6.5.
    - Forget coefficients: $0.10, 0.20$.
    - Retain coefficients: $0.00, 0.00$.
    - Retain set coefficient: 14.51609.
    - Learning rate: $1 \times 10^{-5}$.
    - Batch size: 4.

WMDP 3 folds:

- Stage 1:

    - Activation layer: 7.
    - Layers fine-tuned: $5, 6, 7$.
    - Magnitude: 6.5.
    - Forget coefficients: $0.39, 0.00, 0.00$.
    - Retain coefficients: $0.00, 6.76, 6.76$.
    - Retain set coefficient: 1.
    - Learning rate: $1 \times 10^{-5}$.
    - Batch size: 4.

- Stage 2:

    - Activation layer: 7.
    - Layers fine-tuned: $5, 6, 7$.
    - Magnitude: 6.5.
    - Forget coefficients: $1.00, 4.00, 0.00$.
    - Retain coefficients: $0.00, 0.00, 13.52$.
    - Retain set coefficient: 32.
    - Learning rate: $3 \times 10^{-6}$.
    - Batch size: 4.

- Stage 3:

    - Activation layer: 7.
    - Layers fine-tuned: $5, 6, 7$.
    - Magnitude: 6.5.
    - Forget coefficients: $0.33, 1.33, 5.33$.
    - Retain coefficients: $0.00, 0.00, 0.00$.
    - Retain set coefficient: 45.51609.
    - Learning rate: $3 \times 10^{-6}$.
    - Batch size: 12.

WMDP 4 folds:

- Stage 1:

- – Activation layer: 7.
- – Layers fine-tuned: $5, 6, 7$.
- – Magnitude: 6.5.
- – Forget coefficients: $0.39, 0.00, 0.00, 0.00$.
- – Retain coefficients: $0.00, 10.67, 10.67, 10.67$.
- – Retain set coefficient: 1.
- – Learning rate: $1 \times 10^{-5}$.
- – Batch size: 4.

- Stage 2:
  - – Activation layer: 7.
  - – Layers fine-tuned: $5, 6, 7$.
  - – Magnitude: 6.5.
  - – Forget coefficients: $0.10, 0.20, 0.00, 0.00$.
  - – Retain coefficients: $0.00, 0.00, 16.00, 16.00$.
  - – Retain set coefficient: 8.
  - – Learning rate: $1 \times 10^{-5}$.
  - – Batch size: 4.

- Stage 3:
  - – Activation layer: 7.
  - – Layers fine-tuned: $5, 6, 7$.
  - – Magnitude: 6.5.
  - – Forget coefficients: $0.03, 0.07, 0.13, 0.00$.
  - – Retain coefficients: $0.00, 0.00, 0.00, 32.00$.
  - – Retain set coefficient: 16.
  - – Learning rate: $1 \times 10^{-5}$.
  - – Batch size: 4.

- Stage 4:
  - – Activation layer: 7.
  - – Layers fine-tuned: $5, 6, 7$.
  - – Magnitude: 6.5.
  - – Forget coefficients: $0.03, 0.06, 0.12, 0.25$.
  - – Retain coefficients: $0.00, 0.00, 0.00, 0.00$.
  - – Retain set coefficient: 32.
  - – Learning rate: $1 \times 10^{-5}$.
  - – Batch size: 4.

MMLU 3 folds:

- Stage 1:
  - – Activation layer: 7.
  - – Layers fine-tuned: $5, 6, 7$.
  - – Magnitude: 6.5.
  - – Forget coefficients: $2.00, 0.00, 0.00$.
  - – Retain coefficients: $0.00, 8.00, 8.00$.
  - – Retain set coefficient: 2.

- – Learning rate: $1 \times 10^{-5}$.
- – Batch size: 4.

- Stage 2:

    - – Activation layer: 7.
    - – Layers fine-tuned: $5, 6, 7$.
    - – Magnitude: 6.5.
    - – Forget coefficients: $0.10, 2.00, 0.00$.
    - – Retain coefficients: $0.00, 0.00, 8.00$.
    - – Retain set coefficient: 32.
    - – MMLU retain coefficient: 8.0.
    - – Learning rate: $1 \times 10^{-5}$.
    - – Batch size: 4.

- Stage 3:

    - – Activation layer: 7.
    - – Layers fine-tuned: $5, 6, 7$.
    - – Magnitude: 10.
    - – Forget coefficients: $0.00, 0.07, 1.33$.
    - – Retain coefficients: $0.00, 0.00, 0.00$.
    - – Retain set coefficient: 40.
    - – MMLU retain coefficient: 10.0.
    - – Learning rate: $1 \times 10^{-5}$.
    - – Batch size: 4.

Years 3 folds:

- Stage 1:

    - – Activation layer: 7.
    - – Layers fine-tuned: $5, 6, 7$.
    - – Magnitude: 6.5.
    - – Forget coefficients: $4.00, 0.00, 0.00$.
    - – Retain coefficients: $0.00, 16.00, 16.00$.
    - – Retain set coefficient: 2.
    - – Learning rate: $1 \times 10^{-5}$.
    - – Batch size: 4.

- Stage 2:

    - – Activation layer: 7.
    - – Layers fine-tuned: $5, 6, 7$.
    - – Magnitude: 12.
    - – Forget coefficients: $1.20, 4.00, 0.00$.
    - – Retain coefficients: $0.00, 0.00, 32.00$.
    - – Retain set coefficient: 2.
    - – Learning rate: $1 \times 10^{-5}$.
    - – Batch size: 4.

- Stage 3:

- – Activation layer: 7.
- – Layers fine-tuned: $5, 6, 7$.
- – Magnitude: 12.
- – Forget coefficients: $0.17, 0.67, 2.67$.
- – Retain coefficients: $0.00, 0.00, 0.00$.
- – Retain set coefficient: 36.
- – Learning rate: $1 \times 10^{-5}$.
- – Batch size: 4.

### F.5  SimNPO hyperparameters

WMDP:

- Beta: 0.1.

- Forget coefficients: 8.00.

- Retain coefficients: 1.00.

- Learning rate: $4 \times 10^{-6}$.

- Batch size: 4.

We reran SimNPO with our own implementation as the checkpoint online did not sufficiently unlearn the data on our folds.

### F.6  L-SimNPO hyperparameters

WMDP 3 folds:

- Stage 1:
  - – Beta: 0.1.
  - – Forget coefficients: $3.00, 0.00, 0.00$.
  - – Retain coefficients: $0.00, 1.00, 1.00$.
  - – Retain set coefficient: 4.
  - – Learning rate: $4 \times 10^{-6}$.
  - – Batch size: 4.

- Stage 2:
  - – Beta: 0.1.
  - – Forget coefficients: $0.60, 3.00, 0.00$.
  - – Retain coefficients: $0.00, 0.00, 1.00$.
  - – Retain set coefficient: 6.
  - – Learning rate: $4 \times 10^{-6}$.
  - – Batch size: 4.

- Stage 3:
  - – Beta: 0.1.
  - – Forget coefficients: $5.00, 5.00, 5.00$.
  - – Retain coefficients: $0.00, 0.00, 0.00$.
  - – Retain set coefficient: 6.
  - – Learning rate: $4 \times 10^{-6}$.
  - – Batch size: 4.

