# OpenReview forum: "Layered Unlearning for Adversarial Relearning"
_TMLR — Rejected by TMLR_

### Review · Reviewer_vcna · 2025-10-02

**Summary Of Contributions:**

This paper introduces a novel unlearning algorithm designed to improve the robustness of LLMs against adversarial relearning, where unlearned information is easily recovered by fine-tuning on a small subset of the forgotten data. The authors hypothesize that standard unlearning methods create a single inhibitor mechanism. They address this by sequentially unlearning partitioned data folds to create multiple, distinct inhibitors. The work provides empirical validation in both synthetic and real settings.

**Audience:**

Yes

**Audience Explanation:**

These findings in the paper are interesting but additional information and experiments are required to validate them.

**Claims And Evidence:**

Yes

**Claims Explanation:**

Strengths:

1. The "inhibitor hypothesis” and the link to mechanistic interpretability is interesting.
2. Experiments include both more controlled synthetic and real datasets that is a good way of evaluating the method.

Weaknesses

1. The Layered Unlearning (LU) method involves k sequential stages, resulting in a greater total number of optimization steps compared to a single-shot standard unlearning approach. Could the observed robustness gains be attributed to this increased optimization time rather than the specific layered structure of the algorithm? The paper claims that LU "discovers optima that standard unlearning techniques are unable to discover no matter how long they train". Is there empirical evidence, such as a control experiment where standard unlearning is run for a comparable number of total epochs, to substantiate this claim?
2. The paper evaluates utility preservation by measuring performance on a specified retain set, MMLU. However, this set cannot include all LLM's general knowledge and capabilities, such as fluency and reasoning on out-of-distribution topics. How does the LU method ensure that it does not cause catastrophic forgetting of abilities not explicitly measured by the MMLU retain set? Was any evaluation performed on a held-out set of general knowledge benchmarks, distinct from the one used for regularization during the unlearning process, to validate the preservation of broader model utility? And how does LU compare in this case to unlearning without LU?
3. Table 1/2/3 could you also include the performance on the relearned subsets (and in Table 3 the original performance)? I am wondering whether the increased number of unlearning iterations cause a greater divergence from the initial weights, thereby making it more difficult for the model to relearn the information in the Relearn set?

**Requested Changes:**

Please clarify and provide additional evidence to the "Weaknesses" above.

---

### Review · Reviewer_oH3P · 2025-10-26

**Summary Of Contributions:**

1. The paper proposes a new unlearning schedule that splits the forget set into folds and unlearns them cumulatively and sequentially, while explicitly retaining the remaining folds at each stage.

2. Experiments offer new insight into the limits of post-training behavioral control.

**Audience:**

Yes

**Audience Explanation:**

This work will interest TMLR’s audience because it addresses a core open question in safe and responsible LLM deployment: after we “unlearn” or safety-tune a model, how easily can that ability be restored by downstream fine-tuning or an attacker?

 The paper proposes Layered Unlearning, a general scheduling strategy that can wrap around different unlearning methods and makes adversarial relearning substantially harder, and it evaluates this on real 7B-scale models under realistic attacks.

 This is directly relevant to researchers working on machine unlearning and alignment.

**Broader Impact Concerns:**

This paper is beneficial in safe model deployment.

**Claims And Evidence:**

Yes

**Claims Explanation:**

Authors conduct experiments on both toy models and LLMs to evaluate the proposed method LU combined with RMU and SimNPO baselines. The performance on unlearning and re-training settings shows the effectiveness compared to their counterpart. The experimental results also provide some insights.

**Requested Changes:**

1. Report statistical significance for the claimed improvements of Layered Unlearning (LU) over baselines, not just single-run numbers. The randomisation should come from creation distinct training folds.

2. Please include some additional adversarial relearning settings that reflects realistic misuse.

3. LU requires sequential unlearning over k folds, which the paper acknowledges increases cost and introduces path dependence. It would be interesting to show training time vs. fold number and performance vs. fold number.

---

> ### Author Response · Authors · 2025-11-11
> **Thank you for review**
>
> We thank the reviewer for their comments! Our responses are below:
>
> **R1**: We are currently working on running our experiments to report statistical significance over 3 more distinct training folds. Due to the computational load of running each experiment as described in our general response, we were unable to get them done in the initial two-week period. We have kindly requested an extension in our general response.
>
> **R2**: If the reviewer could elaborate more on what they consider “settings that reflect realistic misuse” that would be helpful. The motivation of this paper is based on a burgeoning issue in open-weight model safety, which is the ability for adversaries to relearn dangerous capabilities simply by finetuning on a small subset of the forget corpus. We are happy to run additional attacks, but are unclear which other settings the reviewer feels are more realistic? For example, we could examine input space attacks such as jailbreaks or latent space attacks presented in [1]
>
> **R3**: We discuss our response to this in our general response as all reviewers had concerns of a similar form.
>
> [1] Che, Zora, et al. "Model tampering attacks enable more rigorous evaluations of llm capabilities." arXiv preprint arXiv:2502.05209 (2025).

---

### Review · Reviewer_Cwnj · 2025-10-27

**Summary Of Contributions:**

This paper proposes a method, called Layered Unlearning (LU), that creates distinct inhibitory mechanisms for a growing subset of the data. Experiments demonstrate the method's effectiveness, but the following issues remain:

1. Abbreviations should be spelled out the first time they appear, such as "SoTA," even though this is a commonly used term. The authors are advised to review the full paper to ensure this is correct.

2. The motivation for this method is unclear. The abstract states, "Recent results suggest that post-training induces shallow context-dependent "circuits" that suppress specific response patterns," but the introduction lacks a corresponding explanation.

3. An intuitive explanation is lacking for why LU is effective.

4. Is the comparison with RMU and SimNPO fair? This is because LU requires $2^k-2$ training steps. Similarly, Figure 1 clearly shows that LU requires more training steps than standard unlearning, which seems ambiguous.

**Audience:**

Yes

**Audience Explanation:**

See Summary Of Contributions.

**Broader Impact Concerns:**

None.

**Claims And Evidence:**

Yes

**Claims Explanation:**

See Summary Of Contributions.

**Requested Changes:**

See Summary Of Contributions.

---

### Decision · Action_Editor_3qAV · 2025-12-09

**Recommendation:** Reject

**Audience:**

Yes

**Audience Explanation:**

This paper studies an important question in safe and responsible LLM deployment about whether the ability of a model can be restored by downstream fine-tuning or an attack. This study is interesting to TMLR audiences. The research should be interesting to people working on machine unlearning and alignment.

**Claims And Evidence:**

No

**Claims Explanation:**

This paper introduces an interesting layered unlearning (LU) algorithm to improve the robustness of LLMs against adversarial relearning. While the paper includes both synthetic and real datasets to evaluate its performance, there are still claims not well supported by accurate evidence. For example, the paper shows a performance gain in robustness. However, all reviewers mention that the proposed algorithm requires much more computational cost than a single-shot standard unlearning approach, and it is not clear whether the robustness gains are attributed to this increased computational cost instead of the layered structure of the algorithm. Moreover, the paper measures its performance on a specific dataset MMLU. It is also not clear whether the LU method will cause catastrophic forgetting of abilities not explicitly measured by the MMLU retain set. Reviewers also mention the necessity to include adversarial relearning settings that reflect realistic misuse.

All three reviewers requested additional experimental results to clarify the performance gains of the proposed LU method. However, there were not enough experimental results added in the discussion phase.